# Comparative Evaluation of the Foot-and-Mouth Disease Virus Permissive LF-BK α_V_β_6_ Cell Line for Senecavirus A Research

**DOI:** 10.3390/v14091875

**Published:** 2022-08-25

**Authors:** Jessica Mason, Victoria Primavera, Lauren Martignette, Benjamin Clark, Jose Barrera, Janine Simmons, William Hurtle, John G. Neilan, Michael Puckette

**Affiliations:** 1SAIC, Plum Island Animal Disease Center, Greenport, NY 11944, USA; 2U.S. Department of Agriculture, Animal and Plant Health Inspection Service, National Veterinary Services Laboratories, Foreign Animal Disease Diagnostic Laboratory, Plum Island Animal Disease Center, Greenport, NY 11944, USA; 3U.S. Department of Homeland Security Science & Technology Directorate, Plum Island Animal Disease Center, Greenport, NY 11944, USA

**Keywords:** Senecavirus A, cell line, swine, LF-BK α_V_β_6_, isolation, FMDV, interferon

## Abstract

Senecavirus A (SVA) is a member of the family *Picornaviridae* and enzootic in domestic swine. SVA can induce vesicular lesions that are clinically indistinguishable from Foot-and-mouth disease, a major cause of global trade barriers and agricultural productivity losses worldwide. The LF-BK α_V_β_6_ cell line is a porcine-derived cell line transformed to stably express an α_V_β_6_ bovine integrin and primarily used for enhanced propagation of Foot-and-mouth disease virus (FMDV). Due to the high biosecurity requirements for working with FMDV, SVA has been considered as a surrogate virus to test and evaluate new technologies and countermeasures. Herein we conducted a series of comparative evaluation in vitro studies between SVA and FMDV using the LF-BK α_V_β_6_ cell line. These include utilization of LF-BK α_V_β_6_ cells for field virus isolation, production of high virus titers, and evaluating serological reactivity and virus susceptibility to porcine type I interferons. These four methodologies utilizing LF-BK α_V_β_6_ cells were applicable to research with SVA and results support the current use of SVA as a surrogate for FMDV.

## 1. Introduction

Senecavirus A (SVA), also identified as Seneca Valley Virus, is the only member of the genus *Senecavirus* and is considered an emerging pathogen in swine. Similar to other animal picornaviruses, SVA possesses a positive sense single stranded RNA genome encapsulated by four viral-encoded structural proteins. The SVA icosahedral capsid is approximately 27 nm in diameter and comprised of 60 protomers consisting of the individual capsid proteins, VP4, VP2, VP3, and VP1, all derived from the P1 polypeptide [1].

Initially isolated and described as a cell culture contaminant [1], SVA is enzootic in swine populations in the United States and other countries [2,3,4]. In the United States, estimated SVA seroprevalence for grower-finisher pigs and sows was 12.2% and 34.0%, respectively [2]. Clinically, SVA is associated with vesicular disease outbreaks and increased neonate mortality [5,6,7,8,9,10,11]. The vesicular lesions on the snout, oral mucosa, hooves, and coronary bands are clinically indistinguishable from those produced by Foot-and-mouth disease virus (FMDV), a high consequence transboundary animal disease pathogen and the causative agent of Foot-and-mouth disease (FMD) [3,6,9,12]. 

The LF-BK α_V_β_6_ cell line is a porcine-derived cell line transformed to stably express the bovine α_V_β_6_ integrin, the primary receptor for FMDV [13,14]. The LF-BK α_V_β_6_ cell line is highly permissive to FMDV and to other swine viruses [15,16]. A subclone of the LF-BK α_V_β_6_ cell line, identified as PIPEC, is used to grow African Swine Fever modified live viruses [17]. In this report we evaluated the LF-BK α_V_β_6_ cell line for use in isolation and propagation of SVA from a field sample submitted for diagnostic investigation of swine vesicular disease to the U.S. Department of Agriculture, Animal and Plant Health Inspection Service, National Veterinary Services Laboratories, Foreign Animal Disease Diagnostic Laboratory. Following SVA virus purification, we adopted and evaluated procedures for usage of the LF-BK α_V_β_6_ cells to determine SVA virus titers, serum neutralizing antibodies to SVA, and sensitivity to type I porcine interferon.

We found that four, widely used methodologies for FMDV research were directly applicable to work with SVA, enabling the potential use of SVA as a surrogate virus to FMDV for preliminary screening of new FMDV countermeasures independent of the high-level biocontainment facilities required for FMDV research.

## 2. Materials and Methods

### 2.1. Origin of SVA Positive Sample and Initial Propagation on Porcine-Derived Cell Lines

Instituto Biologico-Rim Suino-2 (IB-RS-2) cell culture inoculated with a field sample for a diagnostic investigation of swine vesicular lesions, was confirmed as positive for SVA by PCR and kindly provided by the U.S. Department of Agriculture, Animal and Plant Health Inspection Service, National Veterinary Services Laboratories, Foreign Animal Disease Diagnostic Laboratory (FADDL) at the Plum Island Animal Disease Center. Cell culture media was centrifuged at 500× *g* for 5 min and clarified supernatant aliquoted into cryovials for storage at −70 °C.

Porcine-derived cell line LF-BK α_V_β_6_ was grown in DMEM media with 10% fetal bovine serum (FBS) (Hyclone™, Marlborough, MA, USA), 1% minimum essential media non-essential amino acids (MEM NEAA) (Gibco™, Waltham, MA, USA), 1% Sodium Pyruvate (Gibco™, Waltham, MA, USA), and 1% Antibiotic-Antimycotic (Anti-Anti) (Gibco™, Waltham, MA, USA). Porcine-derived PK-15 (ATCC CCL-33) and IB-RS-2 cell lines were grown in DMEM media with 10% FBS (Hyclone™, Marlborough, MA, USA) and 1% Anti-Anti (Gibco™, Waltham, MA, USA). Cells were cultured in T-25 flasks to full confluence and inoculated with 10 μL of the SVA-PCR positive clarified supernatant. Inoculated cells were incubated at 37 °C in 5% CO_2_ until significant cell sloughing was observed, typically, 2–4 days post infection (dpi). Cells were cryofractured at −70 °C, and cellular debris was removed by centrifugation at 500× *g* for 5 min followed by aliquoting and storage of supernatant at −70 °C until further use.

### 2.2. Polyethylene Glycol Precipitation

Cleared cell culture media from inoculated cells was transferred to a 500 mL glass bottle with 80 mg/mL of polyethylene glycol (PEG) and 23 mg/mL of NaCl followed by stirring overnight at 4 °C. After overnight incubation, the solution was centrifuged at approximately 7649× *g* for 25 min using an SS-34 rotor in an RC-6 centrifuge (Thermo Scientific, Waltham, MA, USA). Pellets were resuspended in 3 mL of TNE buffer, pH 7.5, (50 mM Tris-HCl, 100 mM NaCl, 5 mM EDTA), aliquoted, and stored at −70 °C until further use.

### 2.3. Sucrose Density Gradient Centrifugation

Sucrose solutions at 20%, 35%, 42.5%, 50%, and 65% (*w/v*) were prepared in 1X NT buffer (0.15 M NaCl, 0.05 M Tris-HCl), pH 7.8. Sucrose solutions were dispensed into 14 × 95 mm polypropylene tubes (Beckman Coulter, Indianapolis, IN, USA) at decreasing concentrations: 0.8 mL of 65%, 2 mL of 50%, 2 mL of 42.5%, 2 mL of 35%, and 1.8 mL of 20%. To prevent mixing, tubes were stored at −70 °C after the addition of each layer until solid, and boundaries marked. 

One day prior to use, sucrose gradients were thawed at 4 °C. Up to 4 mL of PEG precipitated sample was applied to each gradient and centrifuged at 100,000× *g* for 2 h using a SW50ti rotor in an Optima™ L-70 XP ultracentrifuge (Beckman Coulter^®^, Indianapolis, IN, USA). Individual layers were harvested by pipetting and stored at −70 °C until further use.

### 2.4. Western Blotting

Samples from each gradient layer were diluted to 1X with 4X NuPage™ loading buffer (Invitrogen™, Waltham, MA, USA), heated at 95 °C for 10 min, applied to NuPAGE™ 4–12% Bis-Tris protein gels (Invitrogen™, Waltham, MA, USA) and run according to manufacturer’s instructions. The iBlot™ 2 Dry Blotting System (Life Technologies™, Waltham, MA, USA) was used according to manufacturer’s instructions for transfer to nitrocellulose membranes utilizing iBlot™ 2 Transfer Stacks (Invitrogen™, Waltham, MA, USA).

Membranes were incubated in 5% *w/v* nonfat dry milk blocking buffer solution for 1 h with shaking at room temperature followed by two rinses and a 5-min wash in 1X Phosphate Buffered Saline with 0.05% Tween™ 20 pH 7.4 (PBS-T). Primary antibody incubation was performed overnight at 4 °C with a rabbit anti-SVA VP1 polyclonal antibody (Alpha Diagnostic International, San Antonio, TX, USA) at a 1:1000 dilution followed by three 5-min washes in 1X PBS-T. A 1:500 dilution of goat HRP conjugated anti-rabbit secondary antibody (Thermo Scientific, Waltham, MA, USA), diluted in 1X PBS-T, was applied to the membranes, and incubated for 1 h at room temperature. Membranes were washed three times for 5 min each with PBS-T. To visualize bands, a solution of either 3′,3′-diaminobenzidine, made using SIGMAFAST™ (Sigma-Aldrich, St. Louis, MO, USA) 3,3′-diaminobenzidine tablets (Sigma-Aldrich^®^, St. Louis, MO, USA), or KPL TrueBlue™ (Tacoma, WA, USA) Peroxidase Substrate (Sera care) were prepared as per manufacturer’s instructions and applied to membranes and incubated at room temperature with shaking for 1 h or until bands appeared.

### 2.5. Electron Microscopy

Samples from both the 20% and 35% sucrose fractions of PEG-precipitated LF-BK α_V_β_6_ propagated SVA were evaluated by negative staining on a Hitachi 7600 transmission electron microscope with a 2 k × 2 k AMT camera at 80 kV. Prior to application of sample, 300-mesh formvar/carbon coated grids were inverted on 1% alcian blue for 5 min followed by a rinse in ddH_2_O. Samples were adhered at a 1:5 dilution for 5 min; 2% uranyl acetate was added while blotting across Whatman 50 filter paper.

### 2.6. SVA Plaque Purification and Propagation on LF-BK α_V_β_6_ Cells

The LF-BK α_V_β_6_ cells were cultured to 100% confluence in 6-well plates, rinsed with Dulbecco’s phosphate-buffered saline (DPBS) followed by the application of 2 mL of media containing 2% FBS. The 35% sucrose fractions were diluted from 10^−3^ to 10^−9^ and added to individual wells and incubated at 37 °C in 5% CO_2_ for 1 h with periodic agitation. Media was removed, and cells rinsed with 2 mL of DPBS followed by the addition of 1.5 mL of 10% agar in cell culture media to each well. Plates were incubated at room temperature for 20 min followed by 2 to 3 days at 37 °C with 5% CO_2_ and observed daily for cytopathic effect (CPE). Viruses harvested from individual plaques were inoculated in 6 well plates of LF-BK α_V_β_6_ cells and cultured at 37 °C in 5% CO_2_ for 3–4 days. Supernatant was harvested and evaluated for SVA VP1 by Western blotting as described in Section 2.4.

For serial passaging of selected plaques, inoculated T-25 flasks were harvested 72 to 96 h post infection by cryofracture at −70 °C. Cell culture media was centrifugated at 500× *g* for 5 min and supernatant aliquoted and stored at −70 °C until further use.

### 2.7. Determination of Plaque Forming Units Per mL (PFU/mL)

Plaque assays were performed in 6-well plates on LF-BK α_V_β_6_ cells seeded at 1.2 × 10^6^ cells/well and incubated overnight at 37 °C with 5% CO_2_. Cells were washed with 1 mL DMEM media and inoculated with 0.2 mL/well of 10-fold serial dilutions of supernatant from infected flasks. Plates were incubated at 37 °C for 1 h, rocking plates every 10 min to allow for virus absorption. Media was removed by aspiration and 2 mL of overlay media consisting of 45% of 2X Modified Eagle Medium, 50% of Gum Tragacanth, 2% FBS, 1% NEAA, and 2% Anti-Anti was added to each well and incubated at 37 °C in 5% CO_2_ for 48 h. Following incubation, the overlay media was removed by aspiration; cells were stained with Histochoice Tissue Fixative (Sigma-Aldrich, St. Louis, MO, USA), and plaques were counted for determination of plaque forming units (PFUs) per mL of virus supernatant.

### 2.8. Determination of 50% Tissue Culture Infectious Dose (TCID_50_)

Quantification of TCID_50_ of SVA samples was performed on LF-BK α_V_β_6_ cells in 96-well plates. Supernatant from SVA infected cell culture was diluted from 10^−1^ to 10^−11^ and 100 µL/well dispersed into plates and incubated at 37 °C in 5% CO_2_ for up to 72 h. Wells were scored for CPE under a light microscope, and the titer (TCID_50_/mL), was calculated using the Spearman-Kärber method [18,19].

### 2.9. Sequencing of the Isolated SVA Strain

Sequencing of the VP1 region was performed using primer SVA-1C556F and SVA-2A22R as previously described [20]. For full sequencing, primers were designed from the SVA genome with the highest homology to the VP1 region, GenBank Accession: MH634515, and are listed in Table 1. Based on the genetic uniqueness, the SVA isolate was subsequently referred to as SVA-LP8.

### 2.10. Equilibrium Cesium Chloride Density Gradient Centrifugation

For comparison of particle density between SVA and FMDV by cesium chloride (CsCl) density gradient centrifugation, FMDV A/SAU/26/95 (kindly provided by APHIS FADDL) was cultured in passage 3 BHK-21 cells and inactivated using binary ethylenimine using standard methods. Inactivated virus was PEG precipitated using the same methodology as described in Section 2.2.

SVA-LP8 and FMDV A/SAU/26/95 PEG precipitated viruses was separately layered on 2 mL cesium chloride 2-step discontinuous gradients, 1.38 g/cm^3^ over 1.42 g/cm^3^, prepared in TEN buffer (0.05 M Tris, 0.001 M EDTA, 0.15 M NaCl, pH 7.4). Gradients were centrifuged at 217,485× *g* for 18 h using a SW40Ti rotor in an Optima L-70 XP ultracentrifuge (Beckman Coulter, Indianapolis, IN, USA). Visible bands were collected and dialyzed against PBS at 4 °C using 10K MWCO Slide-A-Lyzer Dialysis Cassettes (Thermo Fisher Scientific, Waltham, MA, USA) prior to Western blotting, as described in Section 2.4.

### 2.11. SVA-LP8 Propagation and Titration in Four Cell Lines

SVA-LP8 was propagated in PK-15, IB-RS-2, LF-BK α_V_β_6_, and BHK-21 cells and virus titers evaluated. Swine derived cell lines, PK-15, IB-RS-2, and LF-BK α_V_β_6_, were cultured as described in Section 2.1, BHK-21 cells were cultured in DMEM media with 10% FBS (Hyclone™, Marlborough, MA, USA), 1% MEM NEAA (Gibco™, Waltham, MA, USA), 1% Sodium Pyruvate (Gibco™, Waltham, MA, USA), 1% L-Glutamine (Gibco™, Waltham, MA, USA) and 1% Anti-Anti (Gibco™, Waltham, MA, USA). Cell cultures in T-25 flasks were infected with 2 mL of SVA at ≈ 10^2.5^ TCID_50_/mL and sampled over 72 h. Infection was performed in technical duplicate for PK-15, IB-RS-2, LF-BK α_V_β_6_ cell lines. SVA titers in the supernatants were determined on LF-BK α_V_β_6_ cells at 24, 48, and 72 hpi and the results averaged together.

### 2.12. Determining Sensitivity of SVA-LP8 and FMDV to Type I Porcine Interferons

Porcine interferon α (IFNα) and interferon β (IFNβ) were produced in transiently transfected cell culture using previously designed plasmids and methodology [21]. LF-BK α_V_β_6_ cells were seeded in 96-well plates at a density of 40,000 cells per well and incubated overnight at 37 °C in 5% CO_2_. After incubation cell culture media was removed, and cells were washed twice using 100 μL of DPBS (Gibco™, Waltham, MA, USA). Dilutions of porcine IFNα and IFNβ aliquots, ranging from 1:100 to 1:409,600, were applied to wells in a total volume of 100 μL, with eight wells utilized per dilution. Plates were incubated at 37 °C in 5% CO_2_ overnight.

After incubation, media was removed and 100 μL per well of infection media (DMEM, 10% FBS, 1% NEAA, 1% L-Glutamine, and 1% Anti-anti) applied containing either FMDV O1 Manisa/69 (3.3 log_10_ TCID_50_/mL) [22], or SVA-LP8 (2.9 log_10_ TCID_50_/mL). Plates were incubated at 37 °C in 5% CO_2_ for 48 h, scoring for CPE at 24 and 48 hpi. 

### 2.13. Determination of Virus Neutralizing Antibody Titers in Swine Serum Samples

Virus neutralizing antibody titers were measured in 96-well plates. Four-fold dilutions of heat inactivated serum samples were mixed with approximately 100 TCID_50_ of either SVA or FMDV per well in duplicate and incubated at 37 °C with 5% CO_2_ for 1 h before adding 2 × 10^4^ LF-BK α_V_β_6_ cells per well and incubating for 72 h. Wells were scored for the presence of CPE under a light microscope and titers calculated by using the Spearman-Kärber method. Titers were expressed as the log_10_ of the dilution that neutralized 50% of the wells (assay lower and upper limits of detection were 0.6 log_10_ and 3.0 log_10_, respectively.

## 3. Results and Discussion

### 3.1. Cultivation of Submitted Field Sample in Porcine Derived Cell Lines

Supernatant from a FADDL vesicular disease investigation that tested negative for FMDV and positive for SVA was inoculated into LF-BK α_V_β_6_, PK-15, and IB-RS-2 cell cultures. At 4 dpi supernatant was applied to discontinuous sucrose gradients. All three gradients demonstrated multiple diffuse bands at different densities suggestive of a potentially diverse viral population within the sample, Figure 1A. Western blotting with a polyclonal antibody against the SVA VP1 protein demonstrated strong reactivity in both the 20% and 35% fractions of LF-BK α_V_β_6_ cultures. PK-15 and IB-RS-2 cell supernatants showed weak reactivity in the 20% fraction only, Figure 1B.

The 20 and 35% sucrose fractions from infected LF-BK α_V_β_6_ supernatants were examined by negative stain transmission electron microscopy. Particles of the expected size for SVA capsids were identified in both fractions, Figure 1C, and the 20% fraction also contained putative empty procapsids, Figure 1D, supporting previous reports of empty procapsids during SVA replication [23].

### 3.2. SVA Purification Using LF-BK α_V_β_6_ Cells

To reduce the carryover risk of lower molecular weight viruses, either porcine or environmental, the 35% sucrose fraction was utilized for plaque purification on LF-BK α_V_β_6_ cells. Cells were evaluated daily for up to four days, and 20 plaques selected. Multiple plaques demonstrated reactivity by Western blot to SVA VP1 polyclonal antibody and supernatant used to continue multiple passages. Subsequently, a single plaque, designated SVA-LP8, was selected for further characterization by genome sequencing.

VP1 sequence results from SVA-LP8 passages 2 and 5 were compared to the original field sample, and no nucleotide changes were detected. A nucleotide BLAST alignment identified a deposited SVA sequence in GenBank, Accession: MH634515, with the highest level of VP1 nucleotide identity to SVA-LP8. Using this sequence primers were designed to allow for sequencing of the SVA-LP8 genome, Table 1. SVA-LP8 genome sequencing identified 6803 nucleotides, including the polypeptide coding region of 6546nt. The consensus nucleotide sequence has been submitted to Genbank, Accession: OP184809. A BLASTN search of NCBI identified strong matches with several other SVA sequences from North America but none with 100% identity. The highest homology, 98.16%, represented SVA USA/IN_Purdue_1581/2016, GenBank: KY618836.1. No sequences representing 100% homology were found from a BLASTp search of the 2181 amino acid polyprotein produced from the SVA-LP8 coding region. The closest homology was to SVA/US/CA/17-60-2D/2017 GenBank: MH634509.1 and contained a 14 amino acid difference.

### 3.3. SVA-LP8 Characterization

SVA-LP8 propagated on LF-BK α_V_β_6_ cells, produced circular plaques with diameters ranging from 3 to 7 mm after 48 h of incubation, Figure 2A, and a titer of 1.9 × 10^10^ PFU/mL. This titer is higher than 10^6^ to 10^8^ PFU/mL titers typically observed with various strains of FMDV [13]. The infected cell culture supernatant was positive for SVA, illustrated by a Western blot of VP1, Figure 2B.

Cultured SVA-LP8 was concentrated by PEG precipitation prior to application on a discontinuous CsCl gradients utilizing the same densities as for FMDV, 1.38 g/cm^3^ and 1.42 g/cm^3^. Application of FMDV produced bands corresponding to infectious virus in the 1.42 g/cm^3^ fraction and empty particles in the 1.38 g/cm^3^ fraction, Figure 3A. SVA-LP8 produced a discrete band containing high titer infectious virus within the 1.38 g/cm^3^ layer, similar to FMDV empty particles, Figure 3A, agreeing with a previously established density of 1.33 g/cm^3^ [24]. The isolated CsCl band was characterized by both silver staining for total proteins and Western blotting for SVA VP1, Figure 3B, confirming the presence of antigen of appropriate size for SVA capsid proteins and specifically the presence of SVA VP1.

### 3.4. SVA-LP8 Propagation in Four Cell Lines

Purification and concentration by CsCl gradient produced a stock of SVA-LP8 used to screen four cell lines for viral propagation. Cells were cultured to confluence in T-25 flasks and infected with CsCl-purified SVA. In addition to the porcine PK-15, IB-RS-2, and LF-BK α_V_β_6_ cell lines, we also evaluated the BHK-21 cell line, previously reported to grow both FMDV and SVA [13,25,26]. While SVA-LP8 grew in all four cell lines, it reached the highest titers within 24 h post infection in LF-BK α_V_β_6_, 9.58 log_10_ to 9.71 log_10_ TCID50/mL, Figure 4. The PK-15 cell line had lower titers, 8.26 log_10_ and 9.15 log_10_ determined at 48 h dpi. IB-RS-2 cells produced the least amount of virus, 6.87 log_10_, over 72-h, Figure 4. These results correlate with observed Western blot results from cell cultures infected with the source field sample, Figure 1B, and confirm that SVA grows rapidly and to a higher titer in LF-BK α_V_β_6_ cells than in other cell lines previously reported for SVA research. Previously published titers obtained from LF-BK α_V_β_6_, BHK-21, and IB-RS-2 cells following FMDV infection correlated with these results [13].

### 3.5. Sensitivity to Type I Porcine Interferons

FMDV is sensitive to both IFNα and IFNβ, [27], and molecular FMD vaccine constructs have incorporated methods to express IFNα to enhance protection [28]. SVA sensitivity to porcine IFNα and IFNβ was evaluated using the LF-BK α_V_β_6_ cell line and compared to that of FMDV O1 Manisa. SVA-LP8 demonstrated a similar sensitivity to both IFNα and IFNβ, Table 2. Interferon antiviral activity against FMDV and SVA was most similar at 48 h post infection. These results raise the possibility that SVA could be used as a FMDV surrogate for screening of cytokines with anti-FMDV potential.

### 3.6. SVA Neutralizing Test Antibody Titers in Serum from Swine Used in FMDV Vaccination Studies

The presence of neutralizing antibodies can be indicative of past infection or vaccination. LF-BK α_V_β_6_ cells can be utilized to determine virus neutralizing antibody titers against FMDV in vaccine studies to evaluate stimulation of an adaptive immune response prior to challenge. Swine serum samples from FMD vaccinated and unvaccinated pigs in previous FMD vaccination studies [29] were screened for antibody titers to SVA both before and after vaccination but prior to challenge. All pigs had been adapted to the facility at least seven days prior to vaccination on Day 0. Serum was screened on study days 0 and 35, Appendix A.

On Day 0, 62% of the population, (53/85), had SVA antibody titers above the 0.6 log_10_ limit of detection, with a mean of 0.93 ± 0.33 log_10_ and a high of 2.4 log_10_, Table 3. This is suggestive of previous exposure to SVA and somewhat expected since SVA is enzootic among swine populations in the U.S. This contrasts with the absence of FMDV antibody titers at Day 0 in all the pigs, Table 3. After 35 days on site, SVA antibody titers increased, remained unchanged or decreased in 42% (36/85), 36% (31/85), and 21% (18/85) of study animals, respectively. Despite these changes, the mean titers of the population, 0.99 ± 0.35 log_10_, remained within the standard deviation for both time points. During this time there was no presentation of clinical signs associated with SVA infection.

When animals were sorted by FMD vaccination status, a small increase in average SVA antibody titers among vaccinated animals was observed at Day 35, Figure 5. However, the difference was not statistically significant when compared to either unvaccinated animals at Day 35 (*p* = 0.54, Student’s paired *t*-test) or to the same animals at Day 0 prior to vaccination (*p* = 0.08, Student’s paired *t*-test). As expected, antibody titers against FMDV were significantly higher in vaccinated animals compared to unvaccinated controls (*p* < 0.001, Student’s paired *t*-test), Figure 5.

While SVA VNTs were determined in this study as an indicator of prior exposure it is unknown if they would indicate protection from virus challenge. In FMD, the presentation of neutralizing antibodies after vaccination is not necessarily indicative of an individuals clinical outcome in swine [29]. These results do validate the need to pre-screen swine for neutralizing antibody titers in SVA experimental studies, as pre-existing antibody levels may continue for a lengthy time after delivery to laboratory facilities.

## 4. Conclusions

In this report, we evaluated the ability of the porcine-derived LF-BK α_V_β_6_ cell line to be used for isolation, culturing, and characterization of a SVA field isolate. We found that SVA grew quickly and to high titers in LF-BK α_V_β_6_ cells. LF-BK α_V_β_6_ cells were useful for SVA isolation and purification, measurement of SVA titers (in both PFU and TCID_50_), the evaluation of the sensitivity to porcine Type I interferon, and the quantification of SVA serum neutralizing antibody titers in swine. The presence of pre-existing antibody titers in swine utilized for FMD and other swine vesicular disease vaccine studies may complicate the interpretation of clinical observations and highlights a need for preliminary screening of pig sera prior to challenge studies.

Due to its high infectivity and devastating economic consequences, research on FMDV is limited to a small number of high-biocontainment facilities. In contrast, SVA can be handled in a BSL2 facility with fewer restrictions. Our work demonstrated that many methodologies used for work with FMDV are directly translatable to SVA. This translatability increases the likelihood that SVA may be useful as a surrogate virus to test and select countermeasures that may also be effective against FMDV.

## Figures and Tables

**Figure 1 viruses-14-01875-f001:**
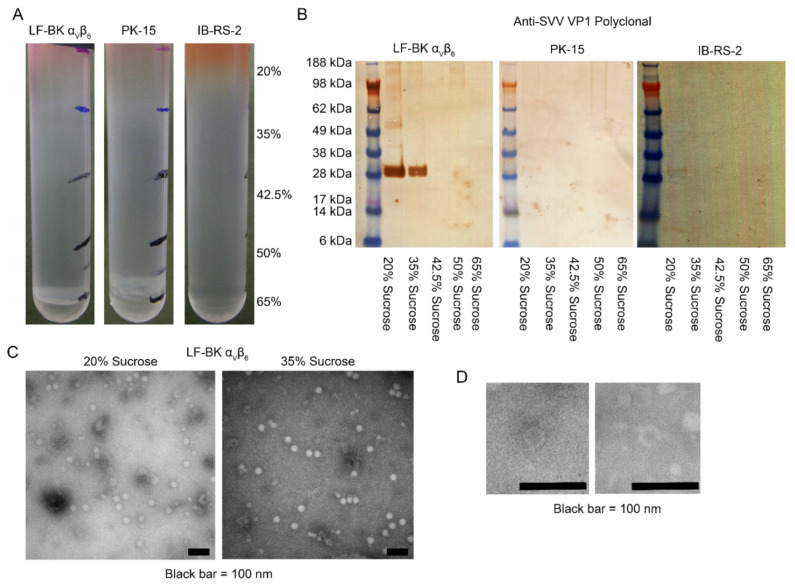
Supernatant from an IB-RS-2 flask inoculated with a submitted field sample cultured in three porcine-derived cell lines, LF-BK α_V_β_6_, PK-15, or IB-RS-2. (**A**) PEG precipitated supernatant was applied to discontinuous sucrose gradients ranging in density from 20% to 65% and demonstrated diffuse materials at all densities. (**B**) Western blotting of harvested gradient layers demonstrated strong SVA antigen in 20% and 35% fractions from LF-BK α_V_β_6_ cultured samples and weak antigen in the 20% fraction for PK-15 and IB-RS-2 cultured samples. (**C**) The 20% and 35% fractions from the LF-BK α_V_β_6_ gradient were examined by transmission electron microscopy using negative staining, and viral particles were observed. Both samples contained particles of the expected size for SVA. (**D**) The 20% fraction also contained structures resembling empty SVA procapsids.

**Figure 2 viruses-14-01875-f002:**
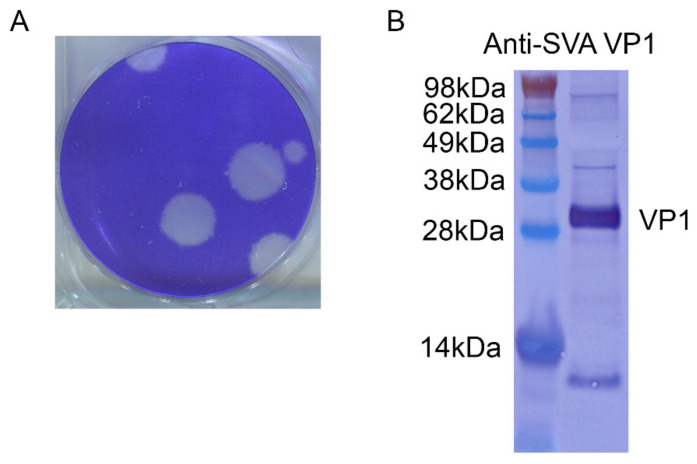
(**A**) Plaque-purified SVA produced 3 to 7 mm plaques on LF-BK α_V_β_6_ cells after 48 h of incubation and (**B**) demonstrated strong reactivity to polyclonal anti-SVA VP1 antibody.

**Figure 3 viruses-14-01875-f003:**
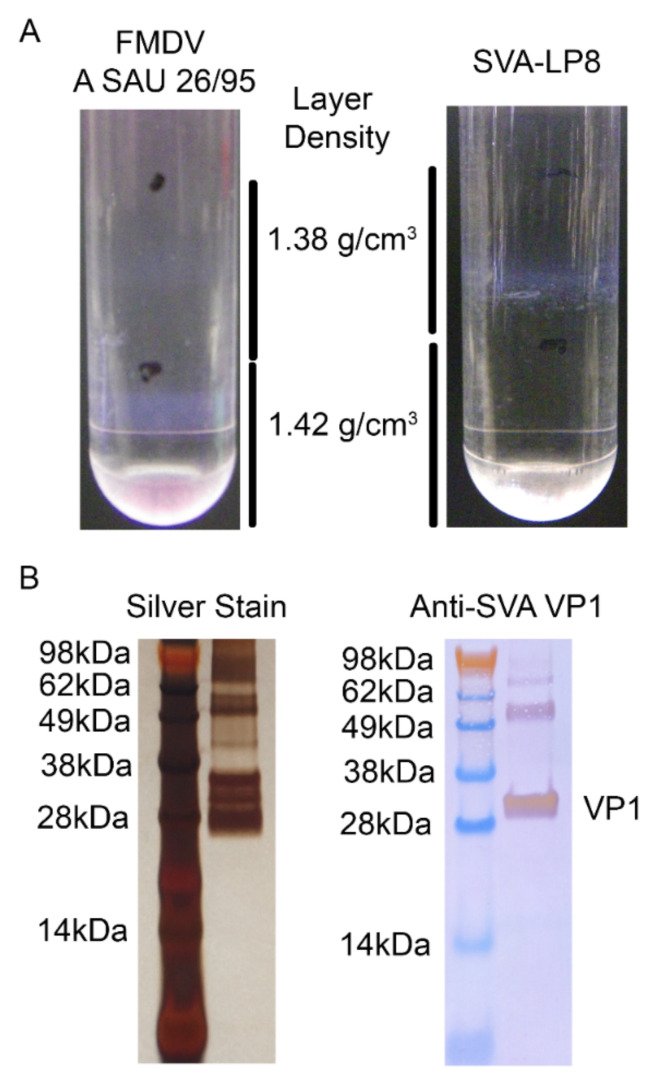
(**A**) Cesium chloride gradient of PEG precipitated supernatant from SVA-LP8 inoculated LF-BK α_V_β_6_ cells produced a discrete white band just above the 1.42 g/cm^3^ density. (**B**) The band harvested from the gradient demonstrated bands of the expected size for individual viral proteins and capsid subunits when run on a denaturing gel and stained. Polyclonal anti-SVA VP1 antibody was reactive with individual VP1 as well as VP1 containing capsid subunits.

**Figure 4 viruses-14-01875-f004:**
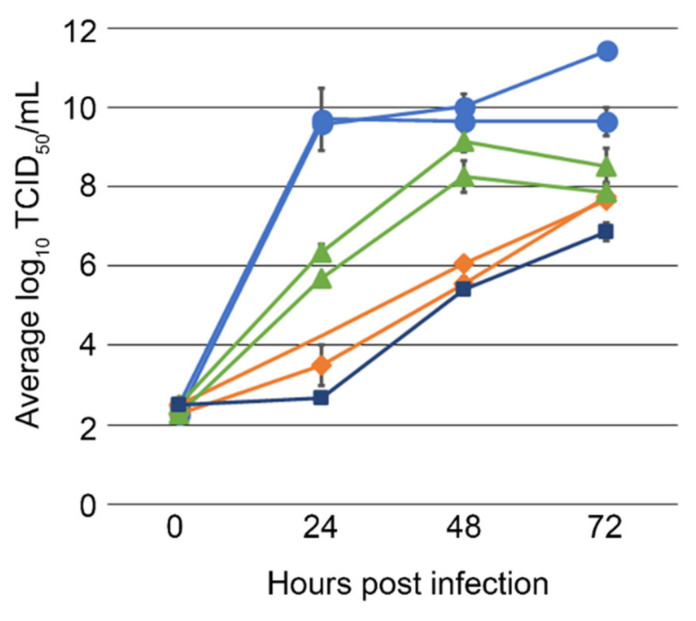
SVA titers, in log_10_ TCID_50_/mL, obtained from LF-BK α_V_β_6_ (blue), PK-15 (green), and BHK-21 (orange), and IB-RS-2 (dark blue) cell lines following infection with ≈2.5 log_10_ TCID_50_/mL of SVA. Titer was determined by the presentation of CPE at 24, 48, and 72 h post inoculation on LF-BK α_V_β_6_ cells and averaged together, the standard deviation is represented by error bars.

**Figure 5 viruses-14-01875-f005:**
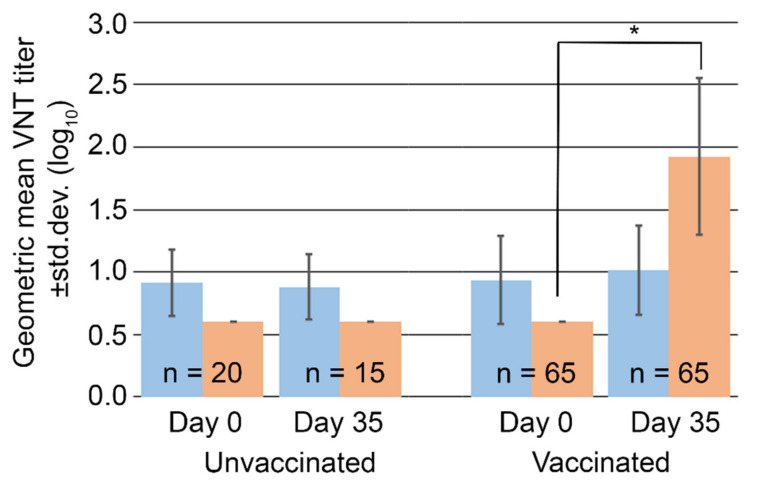
Mean virus neutralizing antibody titers (log_10_) against SVA, blue, and FMDV, orange. No statistically significant differences in SVA titers between unvaccinated and FMD vaccinated swine are observed unlike neutralizing titers against FMD which increased in response to FMD vaccination. * represents *p* < 0.001 using a Student’s paired *t*-test.

**Table 1 viruses-14-01875-t001:** Primers utilized for amplification and sequencing of isolated SVA.

	Forward Primer		Reverse Primer	
	Primer ID	Sequence	Primer ID	Sequence
Amplification	SVF1	GAAGGACTGGGCATGAGGG	SVR550	AGTTCCCGCTGTAGCTCGC
	SVF450	GTGATTGCTACCACCATGAGTACA	SVR1650	CGATAGTATGTACCAAGAGACTGC
	SVF1550	CGAAACCACCCTTGATGTCAAAC	SVR2750	TTAGAGCCAGGAGCCGCCA
	SVF2650	GATTACACCCTCCGTCTCCC	SVR3850	TCCAGTCTTTGACTGTATCCATGG
	SVF3050	GTCACGGTGGTCTCACTGGA	SVR4250	GGCGGAGCTCTGCTTGGC
	SVF4100	ACGACCAGATTGAATACCTCCAGA	SVR5300	TCACCACGGATTGTGAAGCT
	SVF5200	GAGCGAGAATGCTTATGACGG	SVR6400	CGATAGCGGAGCCAAGGAGAA
	SVF6300	GTCTGACCCTGATGTCTTCTGG	SVR7300	TTCTGTTCCGACTGAGTTCTCCCA
Sequencing	SVF1	GAAGGACTGGGCATGAGGG	SVR-SEQ1	ATCCAAGGCACGCTAAGGC
	SVF-SEQ1	CCACCATGAGTACATGGTTCTCC	SVR-SEQ2	TCCGGTAGTCGTCAGACATTTCC
	SVF-SEQ2	CAGTCTCTTGGTACATACTATCGGC	SVR-SEQ3	AGTCTCGGCGTTGTCGGTG
	SVF-SEQ3	ATTGAGGCAGGTAACACTGACAC	SVR-SEQ4	CTGGTGGAGGAGGCGGTTCTA
	SVF-SEQ4	GGCAGTGAGTACCAGGCTTCT	SVR-SEQ5	CTCTGAGGACCACCACAACGG
	SVF-SEQ5	CCAGATTGAATACCTCCAGAACCTC	SVR5300	TCACCACGGATTGTGAAGCT
	SVF-SEQ6	GCGAGAATGCTTATGACGG	SVR-SEQ6	CTATGACGGTCCAGAAGACATC
	SVF-SEQ7	GCCGCCAAGTTTCAATCC	SVR7300	TTCTGTTCCGACTGAGTTCTCCCA

**Table 2 viruses-14-01875-t002:** Sensitivity of FMDV and SVA to type I interferons on LF-BK α_V_β_6_ cells. Interferon activity is presented as units of Interferon Antiviral Activity capable of inhibit 50% of viral growth (IFNAA_50_) per 1 mL after 24 and 48 h of incubation as calculated by the Spearman-Kärber Method.

Cytokine	Virus	Titer (log_10_) TCID_50_/mL	Time (h)	IFNAA_50_ (log_10_) Units/mL
IFNα	FMDV	3.3	24	5.14
			48	4.88
	SVA	2.9	24	5.63
			48	4.84
IFNβ	FMDV	3.3	24	5.48
			48	5.4
	SVA	2.9	24	5.96
			48	5.21

**Table 3 viruses-14-01875-t003:** SVA and FMDV neutralizing antibody titers in swine serum.

Neutralizing Antibody Titers (log_10_)	SVA		FMDV	
	Day 0	Day 35	Day 0	Day 35
0.6	32	25	85	17
0.9	21	19	0	4
1.2	29	28	0	7
1.5	0	4	0	8
1.8	2	3	0	11
2.1	0	1	0	10
2.4	1	0	0	12
2.7	0	0	0	6
3.0	0	0	0	4

## Data Availability

Individual animal SVA and FMDV VNTs at Day 0 and Day 35 along with FMD vaccination status can be found in Appendix A. Any additional data presented in this study is available upon request from the corresponding author.

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
