# Peer review of "Comparative Evaluation of the Foot-and-Mouth Disease Virus Permissive LF-BK αVβ6 Cell Line for Senecavirus A Research"

_viruses, 2022, doi:10.3390/v14091875_

Round 1

Reviewer 1 Report

Article: Comparative evaluation of the Foot-and-Mouth Disease virus permissive LF-BK αVβ6 cell line for Senecavirus A research

This article describes the evaluation of multiple methodologies utilizing LF-BK αVβ6 cells to evaluate Senecavirus A (SVA).  This work is novel as the reviewer is unaware of any other reports evaluating propagation, growth kinetics, virus neutralization, etc. of SVA in LF-BK αVβ6 cells. This study has demonstrated suitability of using this cell line for SVA research and the potential to use SVA in assays as a surrogate for FMDV.

General comments:

1.       Results and Discussion for 3.1 Cultivation in porcine derived cell lines. The reviewer feels that there should be some commentary on the lack of bands observed in the PK-15 and IB-RS-2 cells considering both lines have been used by others performing SVA research. Thoughts on lack of antigen? Very difficult to see even the weak reactivity at 20% suggested in the text.

2.       More information should be provided in the methods or in the figure legion of Figure 4 to help the reader understand why there appears to be two lines for some cell lines, but perhaps only one line for IB-RS-2. In addition, the line appears to be blue in the figure but is listed as black in the figure legend. Also, there is no mention of what the error bars represent. Were multiple flasks titered? Or multiple samples from the same flask at the same timepoints? It is not clear. Again, this Figure seems to demonstrate that SVA grows well in PK-15 cells, so back to point number 1 why do you think you aren’t measuring antigen with your WB?

3.       The methods state 4-fold dilutions utilized for the virus neutralization assay but the titers reported in the table appear to be 2-fold dilutions. How were the upper and lower limits of detection determined for this assay?

·       “A neutralization titer of ≥ 20 would be regarded as neutralization titer-positive.” Liu F, Huang Y, Wang Q, Shan H. Construction of eGFP-Tagged Senecavirus A for Facilitating Virus Neutralization Test and Antiviral Assay. Viruses. 2020 Mar 5;12(3):283. doi: 10.3390/v12030283.

·       “cutoff  VNT  titer  of  64” Goolia M, Vannucci F, Yang M, Patnayak D, Babiuk S, Nfon CK. Validation of a competitive ELISA and a virus neutralization test for the detection and confirmation of antibodies to Senecavirus A in swine sera. Journal of Veterinary Diagnostic Investigation. 2017;29(2):250-253. doi:10.1177/1040638716683214

·       “Serial 2-fold dilutions of serum (1:40 to 1:40,096)” Maggioli MF, Lawson S, de Lima M, Joshi LR, Faccin TC, Bauermann FV, Diel DG. Adaptive Immune Responses following Senecavirus A Infection in Pigs. J Virol. 2018 Jan 17;92(3):e01717-17. doi: 10.1128/JVI.01717-17.

As demonstrated by the above works, many in SVA research would not consider 1:8 or 1:16 as specific neutralizing antibody responses indicative of an SVA neutralizing antibody response. Therefore, the review suggests toning down some conclusions made regarding the neutralizing antibody titers reported. Or acknowledge that these titers may be lower than those reported by others.

4.       Along the lines of the previous comment, the sentence ranging from Lines 329-334 may be misleading the way it is worded. The authors are correct they reported not all animals developed lesions, but animals in both studies had neutralizing titers less than or equal to 16 at the start of the study. The way it is currently written leads to reader to believe that animals were not pre-screened for SVA titers prior to the start of study. The reviewer agrees with the importance of pre-screening animals but this sentence could be structured/worded better to get this point across.

Specific comments

Line 15: the reviewer thinks “typically asymptomatic” wording is too strong based on experimental work and field reports. Suggest something more along the lines of “can be asymptomatic”

Line 32: Don’t need to capitalize picornaviruses

Lines 37-38: Those numbers reported are herd prevalence for that specific study. The reviewer thinks the way the sentence is currently written could be misleading. “The estimated seroprevalence for grower-finisher pigs and sows was 12.2% and 34.0%, respectively.” This sentence from the abstract of that paper may be better to adapt for your goal.

Line 61: Could be helpful to define origin of IB-RS-2 cells

Line 72: replace fetal bovine serum with FBS since it has already been defined

Line 80: Cell doesn’t need to be capitalized

Author Response

  1. Results and Discussion for 3.1 Cultivation in porcine derived cell lines. The reviewer feels that there should be some commentary on the lack of bands observed in the PK-15 and IB-RS-2 cells considering both lines have been used by others performing SVA research. Thoughts on lack of antigen? Very difficult to see even the weak reactivity at 20% suggested in the text.

Response: The images of Figure 1 are from cultures inoculated with the field sample containing SVA and not a purified SVA.  These samples will contain other viral pathogens, both swine and environmental, likely capable of replicating in cell lines.  In discussions with colleagues who work with similar field samples they have noticed that other viruses often out-compete SVA in mixed cultures resulting in the disappearance of SVA from their cultures after 2-4 passages.  This is also supported by the results in Figure 1A which shows diffuse bands at multiple densities. 

As demonstrated in Figure 4, SVA can grow to high titer in the LF-BK αVβ6 cell line within 24 hours however it takes 48-72 hours in PK-15 and IB-RS-2 lines. The authors believe the absence of bands in Figure 1B is the result of SVA being outcompeted in the PK-15 and IB-RS-2 cell lines by other viruses present in the field sample. 

  1. More information should be provided in the methods or in the figure legion of Figure 4 to help the reader understand why there appears to be two lines for some cell lines, but perhaps only one line for IB-RS-2. In addition, the line appears to be blue in the figure but is listed as black in the figure legend. Also, there is no mention of what the error bars represent. Were multiple flasks titered? Or multiple samples from the same flask at the same timepoints? It is not clear. Again, this Figure seems to demonstrate that SVA grows well in PK-15 cells, so back to point number 1 why do you think you aren’t measuring antigen with your WB?

Response: The authors have changed the label of IB-RS-2 from black to dark blue to account for observed difference. 

Samples of media were taken from a single flask over the time course.  For LF-BK αVβ6, PK-15, and BHK-21 cell lines with a similar but slightly different starting virus titer.  Due to its poor performance and its known contamination with a pestivirus we did not repeat the IB-RS-2 growth with the second starting virus titer.

Clarified that the results presented are the average of titers determined at 24, 48, and 72 hours post infection in both materials and methods and Figure 4 caption.  Added that the error bars represent the standard deviation in the caption of Figure 4.

As discussed above, SVA did grow to higher titers in PK-15 over the full time course however it took a longer incubation than with the LF-BK αVβ6 cell line.  As the sample in Figure 1 is a non-purified field sample.  The authors believe that the absence of easily observed bands in for PK-15 in Figure 1B is the result of SVA being out competed by other pathogens present in the field sample.

  1. The methods state 4-fold dilutions utilized for the virus neutralization assay but the titers reported in the table appear to be 2-fold dilutions. How were the upper and lower limits of detection determined for this assay?
  • “A neutralization titer of ≥ 20 would be regarded as neutralization titer-positive.” Liu F, Huang Y, Wang Q, Shan H. Construction of eGFP-Tagged Senecavirus A for Facilitating Virus Neutralization Test and Antiviral Assay. Viruses. 2020 Mar 5;12(3):283. doi: 10.3390/v12030283.
  • “cutoff  VNT  titer  of  64” Goolia M, Vannucci F, Yang M, Patnayak D, Babiuk S, Nfon CK. Validation of a competitive ELISA and a virus neutralization test for the detection and confirmation of antibodies to Senecavirus A in swine sera. Journal of Veterinary Diagnostic Investigation. 2017;29(2):250-253. doi:10.1177/1040638716683214
  • “Serial 2-fold dilutions of serum (1:40 to 1:40,096)”Maggioli MF, Lawson S, de Lima M, Joshi LR, Faccin TC, Bauermann FV, Diel DG. Adaptive Immune Responses following Senecavirus A Infection in Pigs. J Virol. 2018 Jan 17;92(3):e01717-17. doi: 10.1128/JVI.01717-17.

As demonstrated by the above works, many in SVA research would not consider 1:8 or 1:16 as specific neutralizing antibody responses indicative of an SVA neutralizing antibody response. Therefore, the review suggests toning down some conclusions made regarding the neutralizing antibody titers reported. Or acknowledge that these titers may be lower than those reported by others.

Response: Added a section discussing how it is unknown if the VNTs reported in this publication indicate protection for virus challenge.

  1. Along the lines of the previous comment, the sentence ranging from Lines 329-334 may be misleading the way it is worded. The authors are correct they reported not all animals developed lesions, but animals in both studies had neutralizing titers less than or equal to 16 at the start of the study. The way it is currently written leads to reader to believe that animals were not pre-screened for SVA titers prior to the start of study. The reviewer agrees with the importance of pre-screening animals but this sentence could be structured/worded better to get this point across.

Response: The authors have reworded the sentence in question and moved it to the end of the discussion.  The authors have added it into a section discussing addressing previous concerns raised by the reviewer.

Specific comments

Line 15: the reviewer thinks “typically asymptomatic” wording is too strong based on experimental work and field reports. Suggest something more along the lines of “can be asymptomatic”

  • Changed

Line 32: Don’t need to capitalize picornaviruses

  • Changed

Lines 37-38: Those numbers reported are herd prevalence for that specific study. The reviewer thinks the way the sentence is currently written could be misleading. “The estimated seroprevalence for grower-finisher pigs and sows was 12.2% and 34.0%, respectively.” This sentence from the abstract of that paper may be better to adapt for your goal.

  • Changed

Line 61: Could be helpful to define origin of IB-RS-2 cells

  • Added the definition of IB-RS-2 cells

Line 72: replace fetal bovine serum with FBS since it has already been defined

  • Changed

Line 80: Cell doesn’t need to be capitalized

  • Changed

Reviewer 2 Report

In this manuscript, the FMDV permissive cell LF-BK αvβ6 was used to isolate and propagate the SVA from the field sample. Other methodologies that were used for FMDV research including virus titration, evaluation of the sensitivity to porcine type-I IFN, and serum neutralization antibodies against SVA were also performed using LF-BK αvβ6 cells. The results showed that SVA can grow well in LF-BK αvβ cells and these cells are also useful for the other methodologies mentioned before.

The manuscript was well written and the methodologies were logically sound. However, some issues needed to be clarified.

- Isolation and propagation of SVA by using the LF-BK αvβ cell were already demonstrated by one of the reference papers that the authors cited (Gray et al., 2020). With the ability to grow the virus, we can assume from the classical virology class that LF-BK αvβ cells could be used for virus titration and serum neutralization assay as well. Therefore, what is the hypothesis of this study that the authors try to present?

- In the Materials and Methods of "Determining sensitivity of SVA-LP8 and FMDV to type I porcine interferons", the authors did not mention the concentration of IFNα and IFNβ (unit/mL) but came up with the result presented in IFNAA50(log10) Units/mL. So, it is hard to follow where the result came from. Also, how many wells for each dilution that the authors used to calculate the 50% inhibition of viral growth from IFN activity?

- The titers of viruses that were described in Materials and Methods of "Determining sensitivity of SVA-LP8 and FMDV to type I porcine interferons" was not match with the result presented in Table 2. So, please clarify it.

- In the "Determination of virus neutralizing antibody titers in swine serum samples", did the reference sera (known SVA positive and negative) used to validate the assay? How did the authors know that it is accurate? 

- In Figure 5, the addition of the asterisk (*) to show the significant difference in this figure would be helpful.

- In this manuscript, the authors mentioned the idea of using SVA as the surrogate for FMDV research (probably a preliminary test) because of the limitation of the high biosecurity requirement for working with FMDV. However, those two viruses might have the same characteristic in some aspects and also might have different characteristics in some aspects too. In my opinion, it would be nice if the authors could explain more, and perhaps give some examples, about the use of SVA as the surrogate for the FMDV study. 

Author Response

- Isolation and propagation of SVA by using the LF-BK αvβ cell were already demonstrated by one of the reference papers that the authors cited (Gray et al., 2020). With the ability to grow the virus, we can assume from the classical virology class that LF-BK αvβ cells could be used for virus titration and serum neutralization assay as well. Therefore, what is the hypothesis of this study that the authors try to present?

Response:  In Gray et al., 2020, isolation of SVA was obtained directly from homogenized vesicular epithelium tissue.  In this report the virus was obtained after inoculation into a culture of IB-RS-2 cells as required by USDA APHIS FADDL as part of vesicular disease investigations.  This allows for isolation of SVA from samples submitted to USDA APHIS FADDL after the conclusion of investigations. 

- In the Materials and Methods of "Determining sensitivity of SVA-LP8 and FMDV to type I porcine interferons", the authors did not mention the concentration of IFNα and IFNβ (unit/mL) but came up with the result presented in IFNAA50(log10) Units/mL. So, it is hard to follow where the result came from. Also, how many wells for each dilution that the authors used to calculate the 50% inhibition of viral growth from IFN activity?

Response:  Added that eight wells were utilized for each dilution in section 2.12.

The data presented here was determined by evaluating cells treated with multiple dilutions of IFNα and IFNβ samples and infected with a set amount of virus for the presence of CPE.  A unit of interferon activity is defined as the ability to produce a cytopathic effect in 50% of wells. 

- The titers of viruses that were described in Materials and Methods of "Determining sensitivity of SVA-LP8 and FMDV to type I porcine interferons" was not match with the result presented in Table 2. So, please clarify it.

 Response:  The authors inverted the titers from the virus back passages when creating the table, it has been corrected.

- In the "Determination of virus neutralizing antibody titers in swine serum samples", did the reference sera (known SVA positive and negative) used to validate the assay? How did the authors know that it is accurate? 

Response:  These assays were exploratory to screen for the presence of neutralizing antibodies in DHS repository material from FMDV vaccine studies.  There is no set of reference sera available.  The methodology utilizes dilutions of swine sera to determine the point where 50% of wells have inhibition of CPE while utilizing a set amount of virus. 

- In Figure 5, the addition of the asterisk (*) to show the significant difference in this figure would be helpful.

Response:  Added

- In this manuscript, the authors mentioned the idea of using SVA as the surrogate for FMDV research (probably a preliminary test) because of the limitation of the high biosecurity requirement for working with FMDV. However, those two viruses might have the same characteristic in some aspects and also might have different characteristics in some aspects too. In my opinion, it would be nice if the authors could explain more, and perhaps give some examples, about the use of SVA as the surrogate for the FMDV study. 

Response:   The authors bring up the use of SVA as a surrogate for three potential areas.  The first is the screening of anti-viral compounds such as interferons with a particular focus on research that modifies and improves current compounds but might generate variations requiring testing.  The second application is for field testing of disinfection and disposal methodologies.  Testing of these methodologies in field setting is critical for assessing their viability however due to high biosecurity requirements FMDV cannot be utilized.  The final area is to allow for the training of new researchers in methodologies utilized in FMDV research outside of high biocontainment and independent of the need for select agent manipulation authority.

Round 2

Reviewer 2 Report

In this version, the authors revised the manuscript as suggested and all of the questions are clarified.